# Validity and reliability of the German translation of the Diabetes Foot Self-Care Behavior Scale (DFSBS-D)

**Linda Lecker[1], Martin Stevens[2], Florian Thienel[3], Djordje Lazovic[1], Inge van den Akker-Scheek[2], Gesine H. Seeber**[1,2]*

**1** University Hospital for Orthopaedics and Trauma Surgery Pius-Hospital, Medical Campus University of Oldenburg, Oldenburg, Germany, **2** Department of Orthopedics, University of Groningen, University Medical Center Groningen, Groningen, The Netherlands, **3** Diabetes-Center/Endocrinology, Christliches Krankenhaus Quakenbrück, Quakenbrück, Germany

* gesine.seeber@uol.de

## Abstract

### Introduction

Comprehensive regular foot self-care is one of the most critical self-management behaviors for people with diabetes to prevent foot ulcer development and related complications. Yet, adequate foot self-care is only practiced by very few of those affected. To improve diabetic foot syndrome prevention, a valid and reliable instrument for measuring daily foot-care routines in patients with diabetes is needed. However, no such instrument is currently available in the German language. This study, therefore, aims to translate and cross-culturally adapt the "Diabetic Foot Self-Care Behavior Scale" (DFSBS) into German (DFSBS-D) and evaluate its validity and reliability.

### Material and methods

The DFSBS was translated from English into German using a forward-backward procedure as per previous recommendations. Factor analysis was used to study structural validity. To establish construct validity, 21 a priori hypotheses were defined regarding the expected correlation between scores on the new German version (i.e., DFSBS-D) and those of the following questionnaires measuring related constructs: (1) German version "Diabetes Self-Care Activities Measure" (SDSCA-G), (2) "Frankfurter Catalogue of Foot Self-Care" (FCFSP), and (3) "Short Form 36" (SF-36) and tested in 82 patients. To assess test-retest reliability, patients completed the DFSBS-D again after a 2-week interval. Test-retest reliability was assessed from stable patients' data (n = 48) by calculating two-way random-effects absolute agreement ICCs with 95% CI and Bland and Altman analyses. In addition, Cronbach's alpha was calculated as internal consistency measure.

### Results

The 7-item DFSBS-D showed good structural validity. Its single factor explains 57% of the total sample variance. Of the 21 predefined hypotheses, 13 (62%) were confirmed. The

**Data Availability Statement:** Data cannot be shared publicly due to ethical and data protection/privacy reasons. In our informed consent we did not specifically ask participants for permission to

publicly upload their data. We did also not address the possibility of an open/public upload. Thus, participants did not consent to open publication of their data when joining the study. Therefore, a de-identified minimal dataset used and analyzed for this manuscript can only be made available to other researchers on reasonable request by sending an email to the research unit of the University Hospital for Orthopeadics and Trauma Surgery Pius-Hospital, Medical Campus University of Oldenburg (orthopaedie.pius@uni-oldenburg.de).

**Funding:** None of the other authors have any financial disclosure to report.

**Competing interests:** None of the authors have any conflict of interest to report.

DFSBS-D's internal consistency was good (Cronbach's alpha = 0.87). Test-retest reliability over a 2-week interval was also good (ICC 0.76).

## Conclusion

The DFSBS was successfully translated into German. Statistical analyses showed good DFSBS-D structural validity, test-retest reliability, and internal consistency. Yet, construct validity may be debated.

## Introduction

Diabetes mellitus (DM) prevalence has reached epidemic proportions and continues to rise–even in the younger population [1, 2]. Not only has DM a substantial impact on the lives of each affected individual, it also poses considerable socio-economic problems for the whole society [3–5]. Diabetes mellitus is characterized by long-lasting, high blood sugar levels, eventually damaging various organ systems, including the vascular and nervous systems [6]. Diabetic foot syndrome (DFS) is one of the most common DM sequelae [7, 8]. It encompasses various clinical pictures in the patient's foot region and is associated with multiple serious complications [8]. Delayed or ineffective management can even result in amputation of the entire lower limb. In Germany, DM-related amputations are performed every 15 minutes, where up to 70% of patients die within five years following the surgery [9].

To avoid DM-associated foot complications, timely identification of individuals at risk for developing foot problems is of utmost importance [8]. In this context, regular adequate foot self-care is one of the most significant prevention measures [8, 10–15]. However, research shows that adequate foot self-care is practiced far too little [11, 16, 17].

In Germany, a few instruments for assessing diabetic self-care are available [18–20]. However, these instruments are neither easy to administer nor focus on foot self-care. Chin and Huang (2013) developed the 7-item Diabetes Foot Self-Care Behavior Scale (DFSBS) specifically for patients with DFS [16]. The original Taiwanese-Chinese version showed evidence of feasibility, validity, and reliability in the target population. An English translation of the Chinese version is available via the original authors. To use the DFSBS within a German-speaking population of patients with DM, a valid and reliable German version needs to be established. Hence, this study aims to translate and cross-culturally adapt the DFSBS from English into German (DFSBS-D) and subsequently evaluate its psychometric properties.

## Material and methods

The study was conducted between November 2018 and February 2019 at the Diabetes Center Quakenbrück, department of diabetology, metabolic diseases, and endocrinology at the Christliches Krankenhaus Quakenbrück (CKQ), Medical Campus University of Oldenburg and in a specialized private outpatient chiropody clinic in north-western Germany. Before initiation, the study was reviewed and approved by the Medical Ethical Committee of the School of Medicine, Carl von Ossietzky University Oldenburg (2018–063). In addition, the study was registered at the German Clinical Trials Register (DKRS-ID: DRKS00014962).

## Translation and cross-cultural adaptation

The developers of the original DFSBS gave permission to establish a German version of their scale and provided the English version. It was translated into German following a forward-backward procedure as per previous recommendations [21, 22]. First, the original questionnaire was translated into German by two independent translators (T1 and T2) of whom one (T1) was a bilingual resident in internal medicine, while the other (T2) was a German state-approved translator with no medical background. Both translations were synthesized (T12), and a consensus was reached on a preliminary final version. Next, the preliminary German version was tested for wording, phrasing, and understanding in a convenience sample of 12 patients with DM type 1 and type 2. After test-patients' response evaluation, the research team reached a consensus on cultural adaptation and re-phrasing. Next, a non-medical bilingual individual (BT) blinded to the original DFSBS translated the preliminary DFSBS-D (T12) back into English. Finally, the retranslated version was compared to the initial English questionnaire. Comments on the retranslated version were discussed point-wise with all the involved bilingual translators to find a consensus and incorporate final adjustments to the final German DFSBS-D version.

## Participants

Subjects were considered eligible if they (1) were >18 years, (2) suffering from type 1 or 2 DM, and (3) signed written informed consent. Insufficient German language skills or reduced cognitive function to complete the German questionnaire, bilateral leg or foot amputations, and presence of ulcers or wounds precluded participation.

## Procedure

Patients were recruited consecutively in a face-to-face manner during their clinic visits. To test the DFSBS-D's validity and reliability, enrolled patients completed a questionnaire including (1) the newly established DFSBS-D, (2) the German version of the Summary of Diabetes Self-Care Activities Measures (SDSCA-G), (3) the Frankfurter Catalogue of Foot Self-Care-Prevention of the Diabetic Foot Syndrome (FCFSP), and (4) the Short Form Health Survey 36 (SF-36) during their visit. Two weeks after completing the first questionnaire, participants received the DFSBS-D via postal mail and were asked to complete it a second time to determine test-retest reliability. A 2-week interval was considered adequate to assure that clinical change had not occurred [23, 24] and to prevent recall bias. Participants were also requested to evaluate their current state of health concerning their diabetic foot using the simple question: *Have there been any changes in your complaints regarding your diabetic feet compared with 2 weeks ago*? The question had to be rated dichotomously with either *Yes* or *No*. Subjects who reported no changes in complaints were classified as "stable" and their data was used for the reliability analysis [21]. After two weeks, a phone call reminder was used if the re-test questionnaire had not yet been returned.

## Measurement instruments

**Patient and medical characteristics.** The following sociodemographic characteristics were obtained from the self-reported questionnaire: age, sex, height, weight, personal life situation (living alone, with partner and /or children), and educational background (varying from low to high educational levels). Moreover, patients reported the following DM-related information: the number of years diagnosed with DM, type of DM, use of insulin therapy, foot complaints, and kind of foot complaints (e.g., calluses, fissures, or ulcers) if applicable. In

addition, patients were asked to provide information about pre-existing comorbidities in the last six months (e.g., vascular disease or PNP).

**Diabetes Foot Self-Care Behavior Scale (DFSBS).** The DFSBS is a patient self-reported assessment measuring basic foot self-care routines in patients with DM [16]. The original DFSBS was developed by Taiwanese researchers (Chin and Huang) in 2013 and is available in Chinese and English [16]. The 7-item scale has two parts: the first four items (Part 1) relate to certain DM self-care activities and patients are asked about how many days they had executed those in the past week. In the last three items (Part 2), patients are asked to mark the frequency they perform a particular foot self-care behavior. All responses are rated on a 5-point Likert scale. In part 1, possible answers range from 0 days per week (1), 1–2 days per week (2), 3–4 days per week (3), 5–6 days per week (4) to 7 days per week (5). In part 2, answers vary from never (1) to always (5). The DFSBS total score ranges from 7–35, where higher scores represent better foot self-care behavior [16]. With a Cronbach's alpha coefficient of 0.73 and an intraclass correlation coefficient of 0.92 after a two-week interval, the original DFSBS' internal consistency is acceptable, and its test-retest reliability is good. Exploratory factor analysis indicated the DFSBS consists of one factor, explaining 39% of the total sample variance. The DFSBS's construct validity assessment showed a Pearson's correlation of 0.45 between the DFSBS and the subscale foot-care of the Chinese version of the diabetes self-care scale and a Spearman's rho of 0.87 between the DFSBS and the foot-care subscale of The Summary of Diabetes Self-Care Activities Measure [16].

**Summary of Diabetes Self-Care Activities Measure (SDSCA-G).** The SDSCA is a brief multidimensional self-report questionnaire of DM self-management. The latest revised version of the English original was released in 2000 and demonstrated good psychometric properties [18, 19, 25]. A German translation (SDSCA-G) was established in 2013 following previous recommendations [18]. The SDSCA-G demonstrated good reliability and validity in a German cohort of patients with DM type 2 [18]. The SDSCA-G consists of 11 items arranged into five subscales, namely (1) general (2 items) and specific (2 items) diet, (2) exercise (2 items), (3) blood-glucose testing (2 items), (4) foot-care (2 items), and (5) smoking (1 item). A Likert scale ranging from 0–7 is provided for each item, which subjects use to indicate the weekly frequency they perform certain self-care behaviors. The mean number of days is calculated for each subscale (except subscale "smoking") and is used to predict and explore subject's level of self-care [26].

**Frankfurter Catalogue of Foot Self-Care (FCFSP).** The FCFSP, developed by Schmidt et al. in 2005, intends to measure disease-related foot self-care behavior in patients with DM to identify and monitor possible deficiencies [20]. It consists of 19 items describing everyday self-care activities a patient with DM should carry out to prevent DFS [20, 27]. The 19 items are divided into three domains: foot self-control (items 1–9), professional assistance in foot-care (items 10–14), and self-control of shoes and socks (items 15–19) [28]. Each item can be answered on a 5-point Likert scale ranging from never (0), seldom (1), sometimes (2), frequently (3), to always (4). A total score and a score for each subscale can be established [29, 30]. Higher total scores indicate better disease-related foot self-care [27, 29, 30]. Total and/or subscale scores close to zero indicate that a patient needs health care professionals' support to reach adequate daily foot self-care. The FCFSP's test-retest reliability can be considered acceptable [29, 30]. Although different authors state that the FCFSP was valid [27, 29, 30], no specific data is available to our knowledge.

**Short-Form-36-Health Survey (SF-36).** The SF-36 is a self-administered measure to assess subjects' generic health-related quality of life (HLQoL) [31]. The questionnaire comprises 35 items subdivided into eight dimensions: physical functioning (PF, 10 items), role physical (RP, 4 items), bodily pain (BP, 2 items), general health (GH, 5 items), vitality (VT, 4

items), social functioning (SF, 2 items), role emotional (RE, 3 items) and mental health (MH, 5 items) [32, 33]. One additional item, namely retrospective assessment of health-change over one year, cannot be assigned to a specific item subgroup [32, 33]. Response options are dichotomous (i.e., yes/no) or multiple scaled (i.e., 6-dimension Likert scales). Health-related quality of life is reflected by a sum score calculated from the subscales and converted to a 100-point score, with higher scores representing better health status [34]. The German SF-36 is psychometrically robust for data completeness, validity, and reliability within several populations [31, 34–38].

## Statistical analysis

The sample size was chosen to account for a 40% dropout rate and follow the COSMIN guideline. This guideline proposes (1) a minimum number of 100 subjects to assess the internal consistency of health-related patient-reported outcome measures (PROM) and (2) at least 50 subjects of that same sample to evaluate test-retest-reliability [21, 23, 39].

Statistical analysis was performed using IBM®'s Statistical Package for the Social Scientists (SPSS, Version 25; IBM® Corporation, Armonk, NY, USA). Statistical significance was accepted at p≤0.05. Research execution tried to minimize missing values by directly checking each returned questionnaire for data completeness.

**Validity.** Exploratory factor analysis was used to identify DFSBS-D structural validity [40]. We expected that the DFSBS-D was unidimensional similar to the original DFSBS [16]. An eigenvalue ≥1.0 was defined as an extraction criterion, and factor loadings ≥0.40 were considered to represent a high correlation with the respective factor [41].

To establish construct validity, we established 21 a priori hypotheses regarding the magnitude of the relationship between the DFSBS-D and the SDSCA-G, FCFSP, and SF-36 (Table 3). Construct validity was considered good if at least 75% of the predefined hypotheses were confirmed [23, 42]. Spearman´s correlation coefficients ($r_s$) for the between-instruments relationship were calculated and interpreted according to Domholdt (2000): 0.00 to 0.25 very weak, 0.26 to 0.49 weak, 0.50 to 0.69 moderate, 0.70 to 0.89 strong, and 0.90 to 1.00 very strong correlation [43]. The highest correlation was expected between the DFSBS, the SDSCA-G subscale foot-care, and the FCFSP since their items cover the same construct. Correlations of less than 0.26 were expected to exist between the DFSBS and the SF-36 subscales, as they rather assess two different constructs (Table 3).

**Reliability.** Stable subjects' data only were used to establish DFSBS-D reliability. Stable subjects per definition were those patients who reported no change regarding their DM-related foot problems in the re-test questionnaire compared to two weeks earlier. Cronbach's alpha was calculated to investigate DFSBS-D internal consistency. Values between 0.70 and 0.95 were considered indicating good internal consistency [23]. Test-retest reliability was assessed by calculating two-way random-effects, absolute agreement ICCs with a 95% Confidence Interval (CI). An ICC of ≥0.7 is regarded as good test-retest reliability [21, 42, 44, 45]. We expected the resulting ICCs to be ≥0.7 for both DFSBS subscales [44].

Measurement error was analyzed using standard error measurement (SEM) and minimal detectable change (MDC). The former was calculated with the following formula, using the DFSBS-D total scores' pooled SD: $SEM = SD\sqrt{1-ICC}$ [21]. The MDC was calculated on an individual level ($MDC_{ind}$) using the following formula: $MDC\text{ind} = 1.96^*\sqrt{2}^*SEM$, while the MDC on group level ($MDC_{group}$) was calculated by dividing the $MDC_{ind}$ with $\sqrt{n}$ as per previous recommendations [21, 22]. Bland Altman analysis was used to analyze absolute agreement between the first and second DFSBS-D administration. The mean difference of both administrations accompanied by the 95% CI was calculated. Zero lying between 95% CI of the

mean difference was considered absolute agreement, indicating no systematic bias [46]. Limits of agreements (LOA) were defined as the mean difference of both administrations ±1.96 $SD_{(mean\ difference)}$ [46].

# Results

## Demographic characteristics

Overall, 150 patients were invited for participation. Of those, 141 (94%) returned the completed questionnaire. However, fifty-nine questionnaires had to be excluded from the final analyses for following reasons: self-reported ulcers or wounds (n = 54), bilateral foot amputations (n = 2), prediabetes (n = 1), self-reported DM type 3 (n = 1), and refusal to continue participation (n = 1). Thus, complete data of 82 subjects were available for final data analyses. Subjects' demographics are shown in Table 1. The mean age of all participants was 58 ± 15 years, ranging from 20 to 86 years, and 55% were male sex.

Of these 82 subjects, 52 had been asked to fill in the DFSBS-D a second time for test-retest reliability determination. Fifty subjects (96%) returned the questionnaire fully completed. Forty-eight of these subjects (96%) could be classified as "stable" and thus be included for test-retest reliability analysis. Total scores of the questionnaires are displayed in Table 2.

**Table 1. Demographic characteristics.**

| Characteristic | Value* |
|---|---|
| Mean age [years] (n = 82) ** | 58 ± 15 (20–86) |
| Sex (n = 82) | |
| Male | 45 (55%) |
| Female | 37 (45%) |
| Mean BMI [kg/m2] (n = 82) ** | 30.4 ± 6.3 (15.8–49.1) |
| Living arrangements (n = 82) | |
| Alone | 24 (29%) |
| With partner, and/or children | 54 (66%) |
| Other | 4 (5%) |
| Educational level (n = 82) | |
| Elementary school | 37 (45%) |
| Secondary school | 33 (40%) |
| Higher education | 12 (15%) |
| Type of DM (n = 81) | |
| Type 1 | 31 (38%) |
| Type 2 | 50 (62%) |
| Mean number of years diagnosed with DM (n = 80) ** | 15.7 ± 10.9 (0–43) |
| <10 years | 28 (35%) |
| >10 years | 52 (65%) |
| Insulin treated DM (n = 82) | |
| Yes | 66 (80%) |
| No | 16 (20%) |

*Values are n (%) unless otherwise specified.

** Mean ± SD (range)

Abbreviations: BMI, body mass index; DM, Diabetes mellitus; n, number of patients; SD, standard deviation

**Table 2. Questionnaire scores.**

| Scale | Subscale | Value* | SD | Min. | Max. | n |
|---|---|---|---|---|---|---|
| DFSBS-D | | 21.9 | 7.6 | 7 | 35 | 82 |
| SDSCA-G | exercise | 3.3 | 1.9 | 0 | 7 | 82 |
| | blood sugar | 5.2 | 2.6 | 0 | 7 | 82 |
| | foot-care | 2.6 | 2.2 | 0 | 7 | 82 |
| FCFSP | self-control of the feet | 21.5 | 9.1 | 0 | 36 | 80 |
| | professional assistance in foot-care | 13.7 | 7.6 | 0 | 20 | 81 |
| | self-control of shoes and socks | 11.3 | 4.8 | 0 | 20 | 81 |
| SF-36 | PF | 68.4 | 27.5 | 0 | 100 | 82 |
| | RP | 61.0 | 42.1 | 0 | 100 | 82 |
| | BP | 62.0 | 27.7 | 0 | 100 | 82 |
| | GH | 50.4 | 20.1 | 0 | 95 | 82 |
| | VT | 53.4 | 20.7 | 0 | 90 | 82 |
| | SF | 75.8 | 24.6 | 0 | 100 | 82 |
| | RE | 70.3 | 39.2 | 0 | 100 | 82 |
| | MH | 68.4 | 17.9 | 8 | 96 | 82 |

*values are mean or sum scores

Abbreviations: DFSBS-D, German version of the Diabetic Foot Self-care Behavior Scale; FCFSP, Frankfurter Catalogue of Foot Self-Care-Prevention of the Diabetic Foot Syndrome; Max., maximal score; Min., minimal score; n = number of patients; SD, standard deviation; SDSCA-G, German version of the Summary of Diabetes Self-Care Activities Measures; SF-36, Short Form Health Survey 36 (PF, physical functioning; RP, role physical; BP, bodily pain; GH, general health; VT, vitality; SF, social functioning; RE, role emotional; MH, mental health)

## Validity

**Structural validity.** A factor analysis was conducted on the seven DFSBS-D items. The total number of factors was based on the initial eigenvalues. Principal component analysis showed one factor with initial eigenvalues greater than 1.0, explaining a cumulative percentage of 57% of the total variance. Item factor loadings ranged from 0.43 to 0.87.

**Construct validity.** Thirteen (62%) out of 21 predefined hypotheses were confirmed. Spearman correlation coefficients showed correlations ranging from 0.50–0.76 between (items of) the DFSBS-D and items/subscale about foot-care of the SDSCA-G. Correlations between items of the DFSBS-D and FCFSP subscales were lower than hypothesized. The DFSBS-D and the SF-36 subscales showed low correlations as expected. The correlation between the DFSBS-D and the FCFSP subscale self-control of the feet was $r_s = 0.65$, which was higher than the correlation between the DFSBS-D and FCFSP subscale self-control of shoes and socks, confirming hypothesis 21. All calculated correlations between the DFSBS-D and the other scales accompanied by a priori hypothesized values are shown in Table 3.

## Reliability

**Internal consistency.** Cronbach's alpha of the DFSBS-D was 0.87, indicating good internal consistency.

**Test-retest reliability.** Test-retest measures are shown in Table 4. The ICC of the DFSBS-D over a two-week time interval was 0.76 (95% CI: 0.60, 0.87), indicating acceptable reliability. Bland Altman analysis showed that zero was lying outside the 95% CI of the mean difference of both administrations (Fig 1), indicating systematic bias. There was no evidence of proportional bias (B = -.115, p = .283).

**Table 3. Predefined hypotheses and Spearman correlation coefficients.**

| Scales, subscales, and items compared | | Correlation | | Confirmation of hypothesis | p-value |
|---|---|---|---|---|---|
| | | Expected | Spearman | | |
| 1 | DFSBS-D and SDSCA-G subscale foot-care* | >0.70 | 0.76 | Yes | ≤.01 |
| 2 | DFSBS-D item 1 and SDSCA-G item 9 | >0.70 | 0.67 | No | ≤.01 |
| 3 | DFSBS-D item 2 and SDSCA-G item 9 | >0.70 | 0.51 | No | ≤.01 |
| 4 | DFSBS-D item 6 and SDSCA-G item 10 | >0.70 | 0.76 | Yes | ≤.01 |
| 5 | DFSBS-D and FCFSP subscale self-control of the feet** | >0.50 | 0.70 | Yes | ≤.01 |
| 6 | DFSBS-D and FCFSP subscale self-control of shoes and socks*** | >0.50 | 0.62 | Yes | ≤.01 |
| 7 | DFSBS-D item 1 and FCFSP subscale self-control of the feet** | >0.70 | 0.51 | No | ≤.01 |
| 8 | DFSBS-D item 2 and FCFSP subscale self-control of the feet** | >0.70 | 0.47 | No | ≤.01 |
| 9 | DFSBS-D item 6 and FCFSP subscale self-control of shoes and socks*** | >0.70 | 0.58 | No | ≤.01 |
| 10 | DFSBS-D item 1 and FCFSP item 1 | >0.90 | 0.50 | No | ≤.01 |
| 11 | DFSBS-D item 2 and FCFSP item 7 | >0.90 | 0.56 | No | ≤.01 |
| 12 | DFSBS-D item 6 and FCFSP item 15 | >0.90 | 0.77 | No | ≤.01 |
| 13 | DFSBS-D and SF-36 subscale PF | <0.26 | -0.34 | Yes | ≤.01 |
| 14 | DFSBS-D and SF-36 subscale RP | <0.26 | -0.20 | Yes | .06 |
| 15 | DFSBS-D and SF-36 subscale BP | <0.26 | -0.24 | Yes | .03 |
| 16 | DFSBS-D and SF-36 subscale GH | <0.26 | -0.21 | Yes | .06 |
| 17 | DFSBS-D and SF-36 subscale VT | <0.26 | -0.19 | Yes | .08 |
| 18 | DFSBS-D and SF-36 subscale SF | <0.26 | -0.10 | Yes | .37 |
| 19 | DFSBS-D and SF-36 subscale RE | <0.26 | -0.24 | Yes | .03 |
| 20 | DFSBS-D and SF-36 subscale MH | <0.26 | -0.11 | Yes | .33 |
| 21 | The DFSBS-D is expected to correlate higher with FCFSP subscale self-control than FCFSP subscale shoes | Yes | | | |

* SDSCA-G subscale foot-care = SDSCA-G item 9+10;

** FCFSP subscale self-control of the feet = FCFSP item 1–9;

*** FCFSP subscale self-control of shoes and socks = FCFSP item 15–19;

Abbreviations: DFSBS-D, German version of the Diabetic Foot Self-care Behavior Scale; FCFSP, Frankfurter Catalogue of Foot Self-Care–Prevention of the Diabetic Foot Syndrome; SD, standard deviation; SDSCA-G, German version of the Summary of Diabetes Self-Care Activities Measures; SF-36, Short Form Health Survey 36 (PF, physical functioning; RP, role physical; BP, bodily pain; GH, general health; VT, vitality; SF, social functioning; RE, role emotional; MH, mental health)

## Discussion

The DFSBS was successfully translated and cross-culturally adapted into German (DFSBS-D). The results indicate that the DFSBS-D has good structural validity, internal consistency, and test-retest reliability. However, construct validity appears to be questionable.

**Table 4. Reliability measures of the DFSBS-D (n = 48).**

| Measure | Value |
|---|---|
| First administration mean (±SD) | 22.14 (±7.52) |
| Second administration mean (±SD) | 23.56 (±6.79) |
| Mean difference (95% CI) | 1.42 (0.0044, 2.84) |
| ICC (95% CI) | 0.76 (0.60, 0.87) |
| SEM | 2.42 |
| MDC$_{ind}$ | 6.70 |
| MDC$_{group}$ | 0.88 |

Abbreviations: CI, confidence interval; ICC, intraclass correlation coefficient; MDC, minimal detectable change; SD, standard deviation; SEM, standard error of measurement

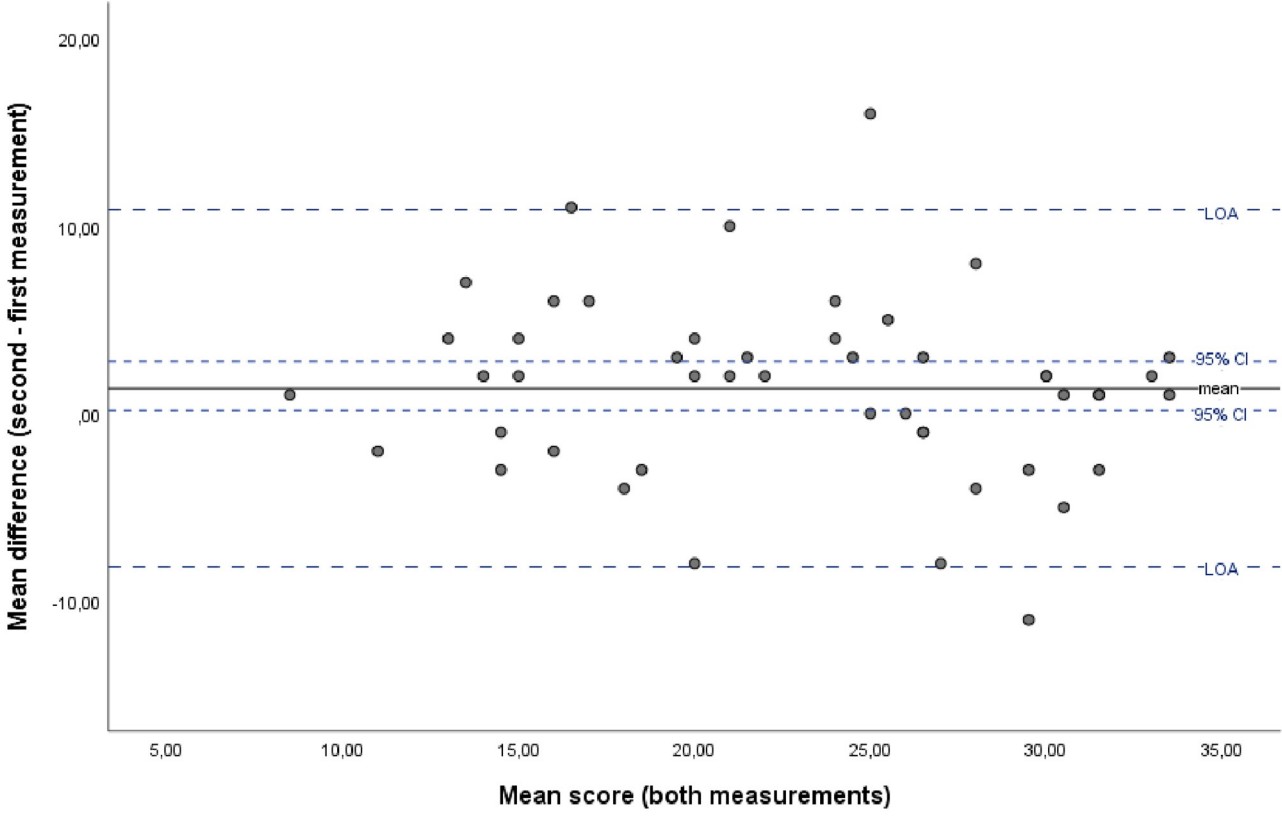

**Fig 1. Bland Altman plot visualizing absolute agreement.** CI = confidence interval; LOA = limits of agreement.

The DFSBS has until now only been available in a Chinese version, which was validated for Taiwan [16]. In addition, an English version was composed from the Chinese questionnaire. However, to our knowledge measurement properties of the English version have not yet been established [16]. Therefore, all results regarding the DFSBS-D found in the present study can only be compared with those of the original Chinese DFSBS version.

The DFSBS-D's structural validity can be considered good. Based on the results of the principal component analysis, the DFSBS-D has a one factor structure, explaining 57% of the total sample variance. While Chin and Huang (2013) determined two factors with eigenvalues greater than 1.0 for the original DFSBS, only one factor was above the scree plot's elbow [16]. Thus, based on the scree plot, the developers set the number of factors to one and considered the original 7-item DFSBS also as unidimensional [16]. The one factor of the original DFSBS explained only 39% of the total sample variance, which is somewhat lower compared to the DFSBS-D. Overall, the results regarding the structural validity of the DFSBS-D are in line with those of the original version.

The construct validity of the DFSBS-D can be considered questionable since only 62% of the 21 predefined hypotheses were confirmed. The COSMIN guidelines recommend that at least 75% of predefined hypotheses must be confirmed to indicate sufficient construct validity [23, 42]. The a-priori set hypotheses concerned the expected correlation between the scores on the DFSBS-D and those of other scales measuring (1) self-care behavior in patients with DM (SDSCA-G), (2) foot self-care behavior in patients with DM (FCFSP), as well as (3) health status and generic health-related quality of life (SF-36). Except for the SDSCA, all other

questionnaires have not been used for hypotheses testing of the DFSBS before. Therefore, the predefined hypotheses of the FCFSP and the SF-36 were theoretically derived and not based on available research findings regarding the DFSBS in other languages. In addition, the lack of information about the validity and reliability of the German-language FCFSP must be considered. This may explain the 38% rejected hypotheses, especially regarding correlations between the DFSBS-D and the FCFSP.

We found a high correlation ($r_s$ = 0.71) between the DFSBS-D and SDSCA-G subscale *foot-care*. That met our expectations as both scales measure a similar construct. Chin and Huang (2013) also indicated a high positive correlation ($r_s$ = 0.87) between the original DFSBS and the SDSCA subscale *foot care* [16]. Consequently, we conclude that, based on correlations between the DFSBS-D and SDSCA subscale *foot-care* and the consistency with previous findings during the validation process of the original DFSBS, the construct of the DFSBS-D is valid.

Regarding the SF-36, it could be speculated that fewer health issues in patients suffering from DM result from better (foot) self-care, leading to greater HRQoL in the respective patient cohort. According to Bonner et al. (2016) and Grady et al. (2011), HRQoL of patients with DM type 2 would increase with the implementation of a more comprehensive self-management education [47, 48]. Nonetheless, since the interdependence of foot self-care and HRQoL is yet unknown, only weak correlations were expected to exist between the DFSBS-D and the SF-36. That was indeed confirmed in the current results, where correlations between the DFSBS-D and all SF-36 subscales were very weak ($r_s$ <0.26).

In general, evaluating measurement properties based on total scores or subscales of an instrument is preferred compared to an item-based evaluation [42]. Yet, as the DFSBS-D has only seven items and no subscales, and following the COSMIN guidelines, at least ten hypotheses should be formulated for validity testing [23], we included six item-based hypotheses. In retrospect, these item-based hypotheses might have been too detailed. On the other hand, they almost scored the expected correlations. Furthermore, they provide relevant information about the measured construct.

The present study results suggest that the DFSBS-D is a reliable patient self-reported instrument. The DFSBS-D's internal consistency (Cronbach's alpha = 0.84) was slightly higher than that of the original DFSBS (Cronbach's alpha = 0.73) [16]. Test-retest reliability of the DFSBS-D (ICC = 0.73) can be interpreted as good [21, 44, 45]. This is in contrast to the original DFSBS, which showed an excellent test-retest reliability (ICC 0.92) over a 2-week interval [16]. However, the ICC is sample-dependent [21]. Thus, the differences in test-retest reliability may be explained by minor differences (e.g., cultural differences) between the current sample and the one of Chin and Huang (2013) [16].

Despite the DFSBS-D's high internal consistency and acceptable test-retest reliability, a systematic bias cannot be entirely ruled out, as witnessed from the Bland and Altman plot (Fig 1). Subjects scored on average 1.78 points higher on the DFSBS-D during the second administration, which might be explained by a better foot self-care behavior initiated by the first DFSBS-D administration.

Self-report measures produce larger measurement errors in general, according to Field (2018), because additional factors influence how people respond [40]. Nevertheless, the relatively high measurement error (SEM 2.54 and $MDC_{ind}$ 7.04) indicates that the DFSBS-D may not be an appropriate instrument to monitor changes between two or more different measurement time points [21, 40]. A minimum difference as high as the $MDC_{ind}$ is required to indicate that an actual change has occurred between two assessments of an individual [21, 40]. This is relatively high considering that the DFSBS-D total score varies between 7–35. Regarding group comparison, a difference of the DFSBS-D mean scores larger than the MDC on group

level (MDC$_{group}$) would speak for an actual between-group difference. However, differences in scores smaller than the SEM cannot rule out measurement error. Therefore, a difference of at least 2.54 is required to detect a statistically significant difference between two DFSBS-D scorings. Since no minimal important change (MIC) values were assessed in this study, it is unclear whether this indicates a clinically relevant difference. To determine the DFSBS-D's MIC further research is required [21].

The number of subjects used for establishing the DFSBS-D's psychometric properties were somewhat smaller (n = 82) than those recommended by the COSMIN panel (n = 100) and initially strived for. However, a subject-to-item ratio of 10:1 is also regarded sufficient for scale validation purposes [45] and as the DFSBS-D consists of seven items, the sample analyzed was still large enough to provide trustworthy results. With respect to test-retest reliability we only missed the targeted number of subjects (n = 50) by two subjects.

## Conclusion

The original DFSBS was successfully translated and culturally adapted into a German version (DFSBS-D). This study's results suggest that the DFSBS-D's psychometric properties are good in terms of structural validity, internal consistency, and test-retest reliability. Construct validity appeared to be questionable at first sight. However, a more in-depth interpretation of the results assumes that the DFSBS-D's construct validity is sufficient. Overall, we conclude that the DFSBS-D is a valid and reliable instrument to assess foot self-care behavior in German-speaking patients with DM type I and II. Future studies are warranted to determine the DFSBS-D's applicability regarding patients with and without PNP and/or POAD, as these are the two main underlying pathologies of the DFS [49].

## Supporting information

**S1 File. DFSBS-D questionnaire.**
(PDF)

## Author Contributions

**Conceptualization:** Martin Stevens, Inge van den Akker-Scheek, Gesine H. Seeber.

**Formal analysis:** Linda Lecker.

**Investigation:** Linda Lecker.

**Methodology:** Martin Stevens, Inge van den Akker-Scheek, Gesine H. Seeber.

**Project administration:** Gesine H. Seeber.

**Resources:** Florian Thienel, Djordje Lazovic, Gesine H. Seeber.

**Supervision:** Djordje Lazovic, Gesine H. Seeber.

**Validation:** Linda Lecker.

**Visualization:** Linda Lecker.

**Writing – original draft:** Linda Lecker.

**Writing – review & editing:** Linda Lecker, Martin Stevens, Florian Thienel, Djordje Lazovic, Inge van den Akker-Scheek, Gesine H. Seeber.

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
