## [Decision Letter · Decision Letter 0]

8 Mar 2022

PONE-D-22-05239Validity and reliability of the German translation of the Diabetes Foot Self-Care Behavior Scale (DFSBS-D)PLOS ONE

Dear Dr. Seeber,

Thank you for submitting your manuscript to PLOS ONE. After careful consideration, we feel that it has merit but does not fully meet PLOS ONE’s publication criteria as it currently stands. Therefore, we invite you to submit a revised version of the manuscript that addresses the points raised during the review process.

We look forward to receiving your revised manuscript.

Kind regards,

Xian-liang Liu

Academic Editor

PLOS ONE

Journal Requirements:

2. PLOS ONE has specific requirements for studies that are presenting a new method or tool as the primary focus, including a newly developed or modified questionnaire or scale (https://journals.plos.org/plosone/s/submission-guidelines#loc-methods-software-databases-and-tools.) One requirement is that the questionnaire or scale must be openly available under a license no more restrictive than CC BY. In light of this, before we proceed, please include a copy of your questionnaire or scale as a Supporting Information file or provide a link if it is available through an online repository. Also, in your Methods section, please discuss whether you obtained the necessary permissions from the owner of the original questionnaire to modify it.

Reviewers' comments:

Reviewer's Responses to Questions

**Comments to the Author**

1. Is the manuscript technically sound, and do the data support the conclusions?

Reviewer #1: Yes

Reviewer #2: Yes

2. Has the statistical analysis been performed appropriately and rigorously? 

Reviewer #1: Yes

Reviewer #2: Yes

3. Have the authors made all data underlying the findings in their manuscript fully available?

Reviewer #1: Yes

Reviewer #2: No

4. Is the manuscript presented in an intelligible fashion and written in standard English?

Reviewer #1: Yes

Reviewer #2: Yes

5. Review Comments to the Author

Reviewer #1: Diabetes foot self-care is a vital part of overall management of both Type 1 and 2 Diabetes. The consequences of foot pathology are extremely serious both to the patient and the health economy. Prevention of foot pathologies, including the vital part played by self-care, are extremely important.

The authors make the point that for diabetes foot syndrome prevention a valid and reliable instrument for measuring daily foot care routines in patients with diabetes is required but no such instruments are available in the German language. Their study aim was to translate and interculturally adapt the Diabetes Foot Self-Care Behaviour Scale (DFSBS) into a German language version (DFSBS-D)and evaluate its validity and reliability. They used accepted and validated methods to translate and validate the DFSBS in the German language and test its reliability. They report the successful translation into German with structural validity, test-retest reliability and internal consistency although they suggest that "construct validity" may be debated. I have just a few minor comments:

1. Their final numbers were less than recommended by COSMIN Guidelines. Do the authors have concerns about this and perhaps this could be explored in their Discussion.

2. Was their any suggestion of a difference in their findings between patients with Type 1 or Type 2 Diabetes or indeed between those on insulin and not on insulin? I realise numbers become relatively small when they try to do this.

3. I'm not keen on their terminology in Table 1 when they refer to "Insulin depended diabetes". There are lots of reasons for people being put on insulin. It would be simpler to use the term "Insulin treated diabetes"

4. It is not considered politically correct these days to talk about "diabetic patients" etc. It is better to use the designation "people/patients with diabetes"-this terminology needs looking at throughout the paper.

5. At no point do the authors clearly state what language they translated from! The original publication was in Chinese. I assume their translation was from the English version. Is that correct? Please add this in the manuscript.

The authors are to be congratulated in producing what appears to be an important development in the diabetes foot self-care management in the German language. My points are minor and overall I was most impressed by this study.

Reviewer #2: The authors have provided a technically sound paper presented in an intelligible fashion assessing the validity and reliability of the German translation of the DFSBS-D. The data do support the conclusions reported in the manuscript and the statistical analyses have been performed appropriately/rigorously with a few comments as described below.

Abstract:

Materials and Methods section: n=82 for the assessment of construct reliability but for test-retest reliability only stable patients n=48 participants assessed (2-weeks between baseline and the 2nd visit). Differences between the “stable” n=48 participants?

Materials and Methods:

• Procedure section: For the test-retest were there differences on anything between those who completed the baseline vs. those who were included in the test-reliability analyses? How many people did not return the questionnaire? Differ3ences on those people as well?

• Statistical Analysis section page 8 line 180-181: what did researchers do with the missingness? Ignore it or some sort of missingness approach?

• Validity Statistical Analysis section: good included interpretation of factor loadings considered high and interpretations of spearman correlation coefficients considered high to weak

• Reliability Statistical Analysis section: were there differences between those who were not included in the test-retest, or those who were not stable as indicated by saying “yes”, any differences in those who did not answer at all ?

Results:

• As mentioned above, were there any differences in the individuals who returned the questionnaire and were categorized as stable/used for the test-retest analyses vs. those who did not say they were stable or did not return the questionnaire?

o If ALL individuals who returned the questionnaires (including those who were not defined as “stable”) did the test-retest results change? Or were they similar?

• Page 13 lines 263-265 (Figure 1). What was the Pearson correlation coefficient for the Bland-Altman analyses? That is the correlation between the differences and means? Was this Pearson correlation coefficient significant? If no you could state there was not proportional biases were observed. Yes, this 0-value is outside the confidence intervals but I would also include the Pearson correlation coefficient/p-value between the differences/means as that will be a good indicator of if there truly was systematic bias.

6. PLOS authors have the option to publish the peer review history of their article (what does this mean?). If published, this will include your full peer review and any attached files.

Reviewer #1: No

Reviewer #2: No

---

## [Author Response · Author response to Decision Letter 0]

4 May 2022

PLOS ONE

Review Manuscript: PONE-D-22-05239

Manuscript Title: Validity and reliability of the German translation of the Diabetes Foot Self-Care Behavior Scale (DFSBS-D)

Oldenburg, 04-06-2022

Re-submission due date: 04-07-2022

Response to Reviewers

Editor’s comment 1: 

Journal Requirements:

2. PLOS ONE has specific requirements for studies that are presenting a new method or tool as the primary focus, including a newly developed or modified questionnaire or scale (https://journals.plos.org/plosone/s/submission-guidelines#loc-methods-software-databases-and-tools.) One requirement is that the questionnaire or scale must be openly available under a license no more restrictive than CC BY. In light of this, before we proceed, please include a copy of your questionnaire or scale as a Supporting Information file or provide a link if it is available through an online repository. Also, in your Methods section, please discuss whether you obtained the necessary permissions from the owner of the original questionnaire to modify it.

Authors’ response: Thank you very much for making us aware about not having followed the correct file naming convention during submission. We now changed this in accordance with the PLOS formatting guidelines. Moreover, we appreciate the journal requires new tools to be publicly accessible. Actually, a copy of the German DFSBS version had already been part of the initial submission as a supplementary file. Nevertheless, we are happy to once again upload a copy together with the revised documents. What we indeed were missing in our initial submission was the original authors official permission to establish a German version of their questionnaire. The e-mail from Dr. Chin is now included in the revised documents as per your request. We also included a sentence in the methods section that the permission to modify the original DFSBS into a German version was obtained from the developer (Lines 79-80)

Authors’ response: We appreciate PLOS One’s policy to provide full open access to the underlying dataset whenever possible. However, after consultation with the responsible data protection officer, publication of our underlying dataset in an open/public repository or supplementary file is not possible due to the strict data protection guidelines in Germany. In our informed consent we did not specifically ask participants for permission to publicly upload their data. We did not even address such a public upload could be a possibility during the publication process. Thus, participants did not consent to open publication of their data when joining the study. From an ethical and data privacy/protection standpoint we should have at least informed the participants about the possibility of an open publication of the study-related database, what we unfortunately failed to do. Therefore, a de-identified minimal dataset used and analyzed for this manuscript can only be made available to other researchers on reasonable request by sending an email to the research unit of the University Hospital for Orthopeadics and Trauma Surgery Pius-Hospital, Medical Campus University of Oldenburg (orthopaedie.pius@uni-oldenburg.de).

Reviewer 1 comments:

General comment: 

Diabetes foot self-care is a vital part of overall management of both Type 1 and 2 Diabetes. The consequences of foot pathology are extremely serious both to the patient and the health economy. Prevention of foot pathologies, including the vital part played by self-care, are extremely important.

The authors make the point that for diabetes foot syndrome prevention a valid and reliable instrument for measuring daily foot care routines in patients with diabetes is required but no such instruments are available in the German language. Their study aim was to translate and interculturally adapt the Diabetes Foot Self-Care Behaviour Scale (DFSBS) into a German language version (DFSBS-D) and evaluate its validity and reliability. They used accepted and validated methods to translate and validate the DFSBS in the German language and test its reliability. They report the successful translation into German with structural validity, test-retest reliability and internal consistency although they suggest that "construct validity" may be debated. I have just a few minor comments:

Reviewer 1 Comment 1

1. Their final numbers were less than recommended by COSMIN Guidelines. Do the authors have concerns about this and perhaps this could be explored in their Discussion.

Authors’ response: We appreciate you are raising this question. Due to the relatively large number of patients that unfortunately had to be excluded due to self-reported foot ulcer and or amputations as outlined in the manuscript, we indeed failed to reach the subject number we initially strived for. However, we think that our analyzed sample still is large enough to provide trustworthy results. In fact, there is an ongoing debate as to how many subjects are required at a minimum for PROM validation (Terwee et al 2007). Next to the suggested minimum of 100 subjects, another accepted rule of thumb is to include a subject-to-item ratio of 10:1 (Nunally & Bernstein 1994). Thus, as the DFSBS presents with only 7 items in total, a minimum of 70 subjects would actually still be sufficient for establishing psychometric properties. Concerning test-re-test reliability we were able to analyze 48 stable subjects. Thus, we missed the targeted number of n=50 by only two subjects, what we think is still acceptable. Nevertheless, we see your point that we should address this aspect in the discussion and included a statement as per your suggestion (Lines 343-348).

References: 

Terwee CB, Bot SD, de Boer MR, et al. Quality criteria were proposed for measurement properties of health status questionnaires. J Clin Epidemiol. 2007;60(1):34-42. 

Nunally JC, Bernstein IH. Psychometric Theory. 3rd ed. New York, NY: McGraw-Hill 1994.

Reviewer 1 Comment 2

2. Was there any suggestion of a difference in their findings between patients with Type 1 or Type 2 Diabetes or indeed between those on insulin and not on insulin? I realize numbers become relatively small when they try to do this.

Authors’ response: Thank you for asking this question. However, we did not investigate into this as indeed our numbers would not have reached the minimal recommended subject-to-item ratio anymore to produce valid and reliable results. The DFSBS-D intents to evaluate daily foot-care routines in terms of a preventive measure in patients with DM. As it is about prevention, one could assume it is not primarily important which type of diabetes one is diagnosed with or whether one is receiving insulin or not as all patients diagnosed with DM should develop a regular foot self-care routine for foot ulcer prevention. Whether there is a difference in adherence to the recommended preventive foot-care depending on the diagnosed type of diabetes or insulin treatment could be evaluated in future studies.

Reviewer 1 Comment 3

3. I'm not keen on their terminology in Table 1 when they refer to "Insulin depended diabetes". There are lots of reasons for people being put on insulin. It would be simpler to use the term "Insulin treated diabetes"

Authors’ response: We appreciate your suggestion and changed the wording in Table 1 accordingly. 

Reviewer 1 Comment 4

4. It is not considered politically correct these days to talk about "diabetic patients" etc. It is better to use the designation "people/patients with diabetes"-this terminology needs looking at throughout the paper.

Authors’ response: Thank you for this comment. Our intend for using the phrase “diabetes patient” instead of “patient with diabetes” was merely to skimp on prepositions and ease reading. However, we appreciate your thought about politically correct language use. Thus, we changed the wording throughout the manuscript as per your suggestion. 

Reviewer 1 Comment 5

5. At no point do the authors clearly state what language they translated from! The original publication was in Chinese. I assume their translation was from the English version. Is that correct? Please add this in the manuscript.

Authors’ response: Thank you for raising this aspect. We indeed translated the English version of the DFSBS, which was provided by the original authors. We added this information to the abstract (Line 30) and the introduction (Lines 66 and 69) as well as to the Material and Methods (Line2 79-80) section. 

Reviewer 1 Comment 6

The authors are to be congratulated in producing what appears to be an important development in the diabetes foot self-care management in the German language. My points are minor and overall I was most impressed by this study.

Authors’ response: Thank you. 

Reviewer 2 comments:

General comment: 

The authors have provided a technically sound paper presented in an intelligible fashion assessing the validity and reliability of the German translation of the DFSBS-D. The data do support the conclusions reported in the manuscript and the statistical analyses have been performed appropriately/rigorously with a few comments as described below.

Reviewer 2 Comment 1

Abstract:

Materials and Methods section: n=82 for the assessment of construct reliability but for test-retest reliability only stable patients n=48 participants assessed (2-weeks between baseline and the 2nd visit). Differences between the “stable” n=48 participants?

Authors’ response: Thank you very much for this question. However, we are not quite sure what exactly you refer to. Data from all 82 eligible subjects were used to establish construct validity (not “construct reliability” as the reviewer mentions). In order to evaluate test-retest reliability we had consecutively send out the DFSBS-D to 52 of the 82 enrolled subjects again, asking them to fill it in a second time. Only data from those patients who answered the question “Have there been any changes in your complaints regarding your diabetic feet compared with 2 weeks ago?” accompanying the re-test questionnaire with “no” were used for the test-retest reliability analysis as per previous recommendations (de Vet et al 2018, Mokkink et al 2010). From n=50 subjects who returned their fully completed re-test questionnaire, n=48 indicated no change and thus got the attribute “stable”. A test-retest reliability analysis is not about how a scale measures a specific construct. Instead, it intends to evaluate the extent to which the scores (here: DFSBS-D scores) for subjects that had not changed over time regarding the measured construct were the same for repeated measures (de Vet et al 2018, Mokkink et al 2010). In that sense, using data from patients who indicate to have changed from the first questionnaire administration to the second does not make sense. We hope this addresses your question sufficiently. Otherwise, we would appreciate to learn more about what exactly you mean with “differences between stable participants”. 

References:

De Vet HCW, Terwee CB, Mokkink LB, et al. Measurement in Medicine: practical guides to biostatistics and epidemiology. Cambridge, UK: Cambridge University Press 2018.

Mokkink LB, Terwee CB, Patrick DL, et al. The COSMIN study reached international consensus on taxonomy, terminology, and definitions of measurement properties for health-related patient-reported outcomes. J Clin Epidemiol. 2010;63(7):737-45. 

Reviewer 2 Comment 2

Materials and Methods:

• Procedure section: For the test-retest were there differences on anything between those who completed the baseline vs. those who were included in the test-reliability analyses? How many people did not return the questionnaire? Differences on those people as well?

Authors’ response: Thank you for this question. We will try to explain that in more detail. As we describe in the methods section, initially 150 patients were consecutively invited for study participation. Of these 141 returned a completed questionnaire. Re-tests were sent out in a consecutive manner to patients who had completed the first DFSBS-D and we stived for at least n=50 stable subjects for the test-retest reliability analysis. Overall, we have sent out 116 consecutive re-tests and n=75 subjects returned them of which n=58 were categorized as “stable”. Thus, we stopped sending out more re-tests at this time point. However, as described in the demographic characteristics section of the manuscript, n=59 subjects had to be excluded leaving n=82 subjects for this study’s analyses. Fifty-two of these n=82 subjects were among the patients that had been provided with a re-test before we stopped sending out re-tests. Thus n=30 out of the n=82 subjects had not gotten a re-test. Response of those who got a retest (n=52) was 100%. However, two (n=2) of the n=52 subjects did not fill in the re-test due to unknown reasons but instead returned it blank, leaving n=50 subjects that returned their re-test questionnaire fully completed. Two (n=2) of these n=50 subjects, however, indicated to have changed over time, thus leaving n=48 “stable” subjects, whose data could be used for the test-retest analysis.

Concerning your question about any differences between those subjects who returned their re-test vs. those who did not, we already mentioned that all 52 subjects returned their re-test. Moreover, there were no significant differences (p>.05) between the n=2 “changed” subjects versus the n=48 “stable” subjects – neither with regard to demographic characteristics nor for the questionnaire scores. In addition, no significant differences (p>.05) could be found between the n=30 subjects who did not receive a re-test anymore versus the n=52 who did. 

• Statistical Analysis section page 8 line 180-181: what did researchers do with the missingness? Ignore it or some sort of missingness approach?

Authors’ response: We appreciate your question. We indeed tried to follow up with the patients about any missing items. However, two patients could not recall when they were first diagnosed with DM and one patient who missed to fill in the diagnosed type of DM was unfortunately not accessible anymore due to incorrect contact data. With regard to the FCFSP only four items in total were missing (two items in subscale “Self-control of the feet” and one item each in the subscales “professional assistance in foot-care” and “self-control of shoes and socks”, respectively). Here, we also tried to follow-up on the missing items. However, in the list of our questionnaires, the FCFSP was the last one to fill in. When asked about the missing items, the respective patients refused to answer these as they felt they had already answered the same/a very similar question in one of the previous questionnaires and thus rejected to answer due to self-perceived redundancy even though we explained the necessity of complete data for each questionnaire. Subsequently, in order to avoid possible distortion of FCFSP subscale results in response to a missing item, we only included subjects with complete data in the respective subscale analysis. 

• Validity Statistical Analysis section: good included interpretation of factor loadings considered high and interpretations of spearman correlation coefficients considered high to weak

Authors’ response: Thank you. 

• Reliability Statistical Analysis section: were there differences between those who were not included in the test-retest, or those who were not stable as indicated by saying “yes”, any differences in those who did not answer at all?

Authors’ response: Thank you for that question. As described in our previous response to the second comment of Reviewer 2 (please see above), re-tests were sent out consecutively and just stopped at a certain time point. There were no statistically significant differences (p>.05) regarding demographic characteristics or questionnaire scores between (1) subjects that were provided with the re-test questionnaire versus those who were not or (2) subjects that were included in the test-retest reliability analysis (i.e., n=48 stable subjects) vs. those who were not (i.e., n=2 changed subjects). The only difference between “stable” versus “changed” subjects was that stable subjects’ foot-related complaints had not changed over time versus a self-reported change over time in foot-related complaints in the “changed” subjects. 

Reviewer 2 Comment 3

Results:

• As mentioned above, were there any differences in the individuals who returned the questionnaire and were categorized as stable/used for the test-retest analyses vs. those who did not say they were stable or did not return the questionnaire?

Authors’ response: Thank you for raising that aspect again. As described in our response to the second comment of Reviewer 2 (please see above), there were no statistically significant differences (p>.05) between the n=2 “changed” subjects versus the n=48 “stable” subjects – neither with regard to demographic characteristics nor for the questionnaires. The only difference between “stable” versus “changed” subjects was that “stable” subjects’ foot-related complaints had not changed during both questionnaire administration timepoints while those of the “changed” patients did. As we were only interested in whether subjects were “stable” or not (i.e., whether their data can be used for test-retest reliability analysis or not) we did not ask for specific details of any change in complaints. Hence, we are not able to report what kind of changes in complaints occurred in the n=2 “changed” subjects as this was beyond the scope of this study.

o If ALL individuals who returned the questionnaires (including those who were not defined as “stable”) did the test-retest results change? Or were they similar?

Authors’ response: We appreciate your question. However, we did not analyze this. The aim of a test-retest reliability testing is to find out about the degree to which a measurement is free from measurement error. Thus, it is about reproducibility or, in other words, it aims at finding out to what extent the score for subjects that had not changed over time were the same for repeated measures (de Vet 2018). Thus, taking data from subjects into account who indicate changes in complaints from the first to the second questionnaire administration is not reasonable. Therefore, for a test-retest reliability analysis one only takes into account results from “stable” patients as described in the manuscript. Otherwise, the calculated coefficient is not valid. 

• Page 13 lines 263-265 (Figure 1). What was the Pearson correlation coefficient for the Bland-Altman analyses? That is the correlation between the differences and means? Was this Pearson correlation coefficient significant? If no you could state there was not proportional biases were observed. Yes, this 0-value is outside the confidence intervals but I would also include the Pearson correlation coefficient/p-value between the differences/means as that will be a good indicator of if there truly was systematic bias.

Authors’ response: Thank you for that suggestion. As the DFSBS-D differences were not statistically different (p>.05), we run a linear regression analysis on the data to verify the absence of proportional bias as per your suggestion. The regression coefficient of the DFSBS-D mean was B=-0.115 with an accompanying insignificant p = .283. This indicates that there is no trend where there are more data points above or below the mean difference line that one would like to see. Thus, bias was truly systematic instead of proportional. We included this information in the results section (Line 264).

---

## [Decision Letter · Decision Letter 1]

20 May 2022

Validity and reliability of the German translation of the Diabetes Foot Self-Care Behavior Scale (DFSBS-D)

PONE-D-22-05239R1

Dear Dr. Seeber,

We’re pleased to inform you that your manuscript has been judged scientifically suitable for publication and will be formally accepted for publication once it meets all outstanding technical requirements.

Kind regards,

Dr Xian-Liang Liu

Academic Editor

PLOS ONE

Additional Editor Comments (optional):

Reviewers' comments:

Reviewer's Responses to Questions

**Comments to the Author**

1. If the authors have adequately addressed your comments raised in a previous round of review and you feel that this manuscript is now acceptable for publication, you may indicate that here to bypass the “Comments to the Author” section, enter your conflict of interest statement in the “Confidential to Editor” section, and submit your "Accept" recommendation.

Reviewer #1: All comments have been addressed

Reviewer #2: All comments have been addressed

2. Is the manuscript technically sound, and do the data support the conclusions?

Reviewer #1: Yes

Reviewer #2: Yes

3. Has the statistical analysis been performed appropriately and rigorously? 

Reviewer #1: Yes

Reviewer #2: Yes

4. Have the authors made all data underlying the findings in their manuscript fully available?

Reviewer #1: Yes

Reviewer #2: Yes

5. Is the manuscript presented in an intelligible fashion and written in standard English?

Reviewer #1: Yes

Reviewer #2: Yes

6. Review Comments to the Author

Reviewer #1: (No Response)

Reviewer #2: All additional concerns and questions have been addressed. The manuscript is much improved compared to the original version.

7. PLOS authors have the option to publish the peer review history of their article (what does this mean?). If published, this will include your full peer review and any attached files.

Reviewer #1: **Yes: **Anthony Howard Barnett

Reviewer #2: No

---

## [Editor Report · Acceptance letter]

24 May 2022

PONE-D-22-05239R1 

Validity and reliability of the German translation of the Diabetes Foot Self-Care Behavior Scale (DFSBS-D) 

Dear Dr. Seeber:

I'm pleased to inform you that your manuscript has been deemed suitable for publication in PLOS ONE. Congratulations! Your manuscript is now with our production department. 

Kind regards, 

on behalf of

Dr. Xian-liang Liu 

Academic Editor

PLOS ONE